# Can Siberian alder N-fixation offset N-loss after severe fire? Quantifying post-fire Siberian alder distribution, growth, and N-fixation in boreal Alaska

**Brian Houseman**[1]☯*, **Roger Ruess**[1]☯, **Teresa Hollingsworth**[2]☯, **Dave Verbyla**[3]‡

**1** Institute of Arctic Biology, University of Alaska, Fairbanks, Alaska, United States of America, **2** Boreal Ecology Cooperative Research Unit, Pacific Northwest Research Station, USDA Forest Service, Fairbanks, Alaska, United States of America, **3** Department of Natural Resources Management, University of Alaska Fairbanks, Fairbanks, Alaska, United States of America

☯ These authors contributed equally to this work.
‡ DV also contributed equally to this work.
* bhouseman@abrinc.com

**Data Availability Statement:** The following DOIs all direct to the underlying data: doi:10.6073/pasta/7f04d011ba2a39b08b794611b54b15ca doi:10.6073/pasta/020191d3b9b72c88a8487b1684eba15c

## Abstract

Fire severity affects both ecosystem N-loss and post-fire N-balance. Climate change is altering the fire regime of interior Alaska, although the effects on Siberian alder (*Alnus viridis* ssp. *fruticosa*) annual N-fixation input (kg N ha$^{-1}$ yr$^{-1}$) and ecosystem N-balance are largely unknown. We established 263 study plots across two burn scars within the Yukon-Tanana Uplands ecoregion of interior Alaska. Siberian alder N-input was quantified by post-fire age, fire severity, and stand type. We modeled the components of Siberian alder N-input using environmental variables and fire severity within and across burn scars and estimated post-fire N-balance using N-loss (volatilized N) and N-gain [biological N-fixation and atmospheric deposition]. Mean nodule-level N-fixation rate was 70% higher 11-years post-fire (12.88 ± 1.18 µmol N g$^{-1}$ hr$^{-1}$) than 40-years post-fire (7.58 ± 0.59 µmol N g$^{-1}$ hr$^{-1}$). Structural equation modeling indicated that fire severity had a negative effect on Siberian alder density, but a positive effect on live nodule biomass (g nodule m$^{-2}$ plant$^{-1}$). Post-fire Siberian alder N-input was highest in 11-year old moderately burned deciduous stands (11.53 ± 0.22 kg N ha$^{-1}$ yr$^{-1}$), and lowest in 11-year old stands that converted from black spruce to deciduous dominance after severe fire (0.06 ± 0.003 kg N ha$^{-1}$ yr$^{-1}$). Over a 138-year fire return interval, N-gains in converted black spruce stands are estimated to offset 15% of volatilized N, whereas N-gains in burned deciduous stands likely exceed volatilized N by an order of magnitude. High Siberian alder density and nodule biomass drives N-input in burned deciduous stands, while low N-fixer density (including Siberian alder) limits N-input in high severity black spruce stands not underlain by permafrost. A severe fire regime that converts black spruce stands to deciduous dominance without alder recruitment may induce progressive N-losses which alter boreal forest ecosystem patterns and processes.

doi:10.6073/pasta/0600e58dea74dd5df84153af35da6f56 doi:10.6073/pasta/4694dfc87d322b4891a12e3c3fdabe18 doi:10.6073/pasta/a8acc1f2107944a9e9ad91974c7aff91 doi:10.6073/pasta/a5eed7ac3329a4aca7fb6e54185f0c50 doi:10.6073/pasta/705056665d58f1138d9707f262487482 doi:10.6073/pasta/f0b1cb6152f360c42a11db3d5cd2a61a doi:10.6073/pasta/7496cd1d2f43929266e7feca934d1621 doi:10.6073/pasta/37703e3d1a3aa6c41a491cbba91a3ce1.

**Funding:** This research was supported in part by the Bonanza Creek Long-Term Ecological Research Program which is funded by the National Science Foundation (award number DEB-1636476) (RR, TH); the USDA Forest Service, Pacific Northwest Research Station (RJVA-PNW-01-JV-11261952-231) (TH); and a student research grant from the UAF Center for Global Change & Arctic System Research (15-010) (BH). The funders had no role in study design, data collection and analysis, decision to publish, or preparation of the manuscript.

**Competing interests:** The authors have declared that no competing interests exist.

## Introduction

Alder (*Alnus* spp.) forms a symbiotic relationship with the nitrogen-fixing *Frankia* bacteria resulting in significant implications for N cycling in post-disturbance ecosystems throughout interior Alaska and elsewhere in the boreal forest [1–4]. Annual stand-level alder N-fixation input can exceed 100 kg N ha$^{-1}$ yr$^{-1}$, leading to substantial increases to ecosystem N pools in primary succession on floodplains [4,5] and recently deglaciated uplands [6–8], as well as some post-fire stand types of boreal forest uplands [2,4,9]. Alder N-fixation input is associated with altered biogeochemical patterns such as soil acidification, increased N cycling and availability, and elevated aquatic productivity [10–12]. However, the ecosystem consequences of Siberian alder (*A. viridis* ssp. *fruticosa*) N-fixation input after fire disturbance in black spruce (*Picea mariana*) forests of Alaska are poorly understood.

Climate warming in interior Alaska over the past 60 years [13] has increased the frequency, size, and severity of wildfires [14]. This change in fire regime is linked to a shift in dominant vegetation from black spruce to deciduous-dominated forests, representing a vegetation transition that is novel over the past several thousand years [15,16]. During the extreme 2004 Alaska fire season, volatilization of soil organic N ranged from 0–94% and averaged 50% in burned black spruce stands [17]. Sustained losses of N—resulting from higher rates of N-volatilization than N-fixation input—have been observed in the fire dependent longleaf pine savannas of the southeastern United States [18]. Despite the importance of fire in shaping community and ecosystem dynamics in the boreal forest [19], little is known about the effect of fire on the density, growth, and N-fixation of Siberian alder (interchangeably referred to here as alder) or the associated feedbacks to ecosystem N balance and post-fire plant community development. The historic boreal forest fire regime (i.e., predominantly low to moderate severity fires) facilitated increased alder growth and reproduction on burned areas in Canadian boreal forest [20]. However, evidence from a Swedish boreal forest suggests that during extreme fire events, there is a complete destruction of rhizomatous shrubs and seed banks [21]. Extreme fire events within the Alaskan boreal forest could significantly reduce alder recolonization and resprouting, thereby limiting total N inputs and subsequently ecosystem resilience to disturbance. Post-fire alder development is influenced by pre-fire alder distribution, as well as the patterns and factors affecting alder recruitment, growth and N-fixation across upland boreal forest stand types (e.g., black spruce versus deciduous dominance). While it is likely that these factors are strongly influenced by fire severity, the interactive effects of fire severity and site conditions on alder density, growth and N-fixation have not been studied.

Alder has been described as a common component of post-fire successional dynamics [22], yet its abundance in various post-fire successional stands ranges from absent to very dense [23,24]. Such a variable distribution is likely due to a combination of fire history and other environmental factors. Disentangling the effects of Alaska's changing fire regime on alder distribution and abundance and the associated impacts on post-fire N-balance and ecosystem resilience requires examination of the patterns and factors influencing alder N-fixation inputs and their relationship to fire severity effects at landscape scales. Because high fire severity has been shown to reduce post-fire rhizomatous shrub abundance in the boreal forest [21], we hypothesized that high severity fires limit post-fire alder density and therefore stand-level N-fixation inputs during secondary succession. However, in order to test for an effect of fire severity on alder density (and therefore stand-level N-fixation input), we must disentangle fire severity effects from other potential effects. Our specific objective was to characterize how alder density, nodule-level N-fixation, nodule-level biomass, alder ramet and leaf traits, and plant-level N-fixation input vary across a fire severity gradient, fire age, and environmental characteristics (soil and topography). Our final goal of this study was to describe the

relationship among fire severity, alder symbiotic N-fixation input, and post-fire N-balance. Our data show that high severity fire substantially reduces alder density which contributes, at least initially, to a strong imbalance between the high amount of N that was volatilized and the low amount of N-input during secondary succession.

## Materials and methods

### Study area

Our study area encompassed two burn scars within the Yukon-Tanana Uplands ecoregion [25] of interior Alaska. Study plots were located on public land owned by the State of Alaska and the Fairbanks North-Star Borough, and private land owned by the University of Alaska Fairbanks. Access to University of Alaska Fairbanks lands was granted by the Poker Flats Research Range and the Bonanza Creek Long Term Ecological Research program. This field study did not involve protected or endangered species. One of the burn scars sampled—the 1971 Wickersham Dome Fire (WDF)—is located approximately 35 km NW of Fairbanks, Alaska (64.9˚ N, 147.9˚ W) and covers 5,500 ha. The 2004 Boundary Fire (BF) scar is located approximately 40 km NE of Fairbanks and covers over 210,000 ha. Study plots were located between 100 m and 5 km from the burn scar edge and did not overlap roads, trails, fire-fighting treatments or other unnatural features. Fire history records indicate that no other fires occurred in the study area at least since 1940 [26]. Discontinuous permafrost is found 40–50 cm below the soil surface, but ridgetops and upper south-facing slopes often lack permafrost [27]. Throughout the study area alder occurs as a tall shrub that can form dense patches with multiple individuals close to one another—particularly on disturbed sites such as trails and roadsides where the mineral horizon is exposed.

### Field and laboratory methods

**Siberian alder density.** Siberian alder density (plants ha$^{-1}$) sampling occurred in summer 2014. Each study plot was circular (100-m diameter) and plots were spaced 200 m apart along randomly located toposequence transects of varying lengths [WDF (n = 21 transects), BF (n = 40 transects)]. Plots within the BF were randomly located within the same topographic range sampled in the WDF (324–581 m elevation, 0–360˚ aspect, and 0–26˚ slope). Alder density was estimated by measuring the distance between plot center and the nearest alder (if >50 m, then truncated to 50 m) on WDF sample plots (n = 80) and BF sample plots (n = 183). An individual alder was defined as one or more tightly clustered ramets spatially distinct from another individual on the same plot. Distances were later used in a non-parametric formula for estimating plant density (see Statistical analysis). We chose to measure alder density with this method because it is a more robust estimator of non-parametric plant distributions than other methods of estimating plant density [28–30].

**Nodule-level N-fixation.** A subsample of the 2014 alder density plots was chosen for N-fixation sampling between June 29th and July 29th of 2015—the peak of the season for N-fixation activity [4,31]. The 2014 plots were stratified by burn scar and stand type (see *Stand types, topography, and fire severity*). We alternated sampling between burn scars daily so that sampling dates for each burn scar were evenly spread throughout the 30-day sampling period. On each day we randomly selected a stand type and then randomly selected a plot within the chosen stand type for measurement. This method of selecting plots controlled for variation in N-fixation activity throughout the day and throughout the 30-day sampling period. We subsampled as many of the 2014 plots as possible during the 30-day sampling period. Within the BF, nodules from 48 alders across 19 plots and four stand types were sampled for N-fixation, and in the WDF, nodules from 48 alders across 21 plots and three stand types were sampled. At

each plot, experimental or control nodules were collected from alders of representative height and vigor. Experimental nodules were then incubated in $^{15}N_2$ gas following previously established methods for alder nodule $^{15}N_2$ uptake [31]. Experimental and control nodules from individual alders were separately stored in 5 ml cryovials which were preserved in liquid nitrogen during transfer from the field to the lab. In the lab, nodules were dried at 65 ˚C and ground before mass spectrometry analysis [31–33]. Calculations of nodule-level N-fixation rate (μmole N g nodule$^{-1}$ hr$^{-1}$) followed previous methods that account for the fixation of both $^{15}N_2$ and $^{14}N_2$ [4,31,32]. Rates of nodule-level N-fixation in this study reflect maximal rates relative to the assumed lower rates in spring, fall and other periods of summer.

**Nodule-level biomass.** Between July 30$^{th}$ and September 28$^{th}$, 2015, plant-level live nodule biomass (g nodule m$^{-2}$ plant$^{-1}$) was sampled at each plot that had been sampled for N-fixation [BF (n = 19), WDF (n = 21)]. At each plot, we systematically selected five alders of representative height and vigor that were not previously sampled for nodule-level N-fixation rate [BF (n = 95 alders), WDF (n = 105 alders)]. Below each alder five random soil cores (5.5 cm diameter) were collected within the area of nodulation—an area we defined as a 1 m buffer around the perimeter of the outermost ramets of an individual alder. Each soil core included the entire organic horizon and the upper 5 cm of mineral horizon. Cores from a single alder were pooled and plant-level live nodule biomass was calculated in the lab using established methods [32,33]. The plant-level live nodule biomass values were then averaged at each plot for a plot-level estimate of plant-level live nodule biomass.

**Siberian alder ramet and leaf traits.** For each alder density plot sampled in 2014 (n = 80 plots in the WDF, n = 183 plots in the BF), the plot was divided into four quarters; in each quarter the ramet basal diameter (cm), ramet height (m), number of ramets, and live or dead status of each ramet were measured for the individual alder nearest to plot center. For each alder that was sampled for nodule-level N-fixation in 2015 (n = 48 alders across 21 plots in the WDF; n = 48 alders across 19 plots in the BF) 10 leaves were collected at the time of N-fixation sampling and used to estimate specific leaf mass (mg cm$^{-2}$) following Ruess and others [33].

**Stand types, topography, and fire severity.** For each alder density plot sampled in 2014 (n = 263), dominant plant cover was quantified using the Braun-Blanquet relevé method [34]. Trees and shrubs were identified to species, grasses and forbs to genus, and all non-vascular plants as *Sphagnum*, "other moss," or "lichen." Cover was determined separately for dead trees (diameter > 7.6 cm) that were on the ground and all other forms of plant litter. Post-fire stand types of each burn scar were determined with a hierarchical clustering of plot-level cover estimates using PC-ORD, Version 5.0; Euclidean distance measures and Warde's linkage method were used in the calculation [35]. Any species occurring in < 5% of all plots was removed from the dataset before relativizing species cover based on each species maximum cover. Indicator Species Analysis (ISA) in PC-ORD [36] was used to identify which hierarchical grouping configuration produced the most distinct stand types for each burn scar. Pre-fire stand types for the BF were determined by the proportion of burned tree species (standing and down) and unburned canopy-dominant tree species. Pre-fire stand types were later confirmed using a pre-fire satellite image (May 2002 Landsat 7 ETM image).

Sample plot slope, aspect, and elevation, and solar radiation were determined using a 30-m resolution digital elevation model and the Spatial Analyst toolbox in ArcGIS [37]. Fire severity was determined for each plot within the BF using the difference in normalized burn ratio (dNBR) as derived from 30-m resolution Landsat imagery [38]. Raster pixels with a majority overlap on a circular sample plot were averaged together for a plot-level mean dNBR value. Fire severity was not determined for the WDF plots due to a lack of pre-fire satellite imagery.

**Soil properties.** Beneath each alder sampled for nodule-level N-fixation (n = 48 alders across 21 plots in the WDF; n = 48 alders across 19 plots in the BF), moisture (%) and

temperature (˚C) of the organic horizon were measured with a CS620 HydroSense water-content probe (Campbell Scientific, Logan, UT, USA) and a TM99A REOTEMP digital thermometer (REOTEMP Instruments, San Diego, CA, USA), respectively. We controlled for variation in temperature and moisture among stand types and between burn scars through the previously described method of random plot selection during the 30-day sampling period.

Additional soil measurements were collected on plots sampled for nodule-level N-fixation (n = 21 plots in the WDF; n = 19 plots in the BF), but this time measurements were collected around a randomly chosen alder of representative height and vigor that was not sampled for N-fixation. Depth of organic horizon (cm) was measured at ten random locations along a randomly oriented, 20 m transect that bisected the alder at 10 m. Of the ten soil depth locations, one was randomly selected for Oi (fibric—minimally decomposed organic matter), Oe (hemic—moderately decomposed organic matter), and Oa (sapric—highly decomposed organic matter) depth measurements and a mineral horizon sample approximately 5 cm below the organic horizon. Bulk density (g cm$^{-3}$), soil pH, total nitrogen (% N), carbon (% C) and phosphorus (% P) were all measured from mineral horizon samples following Mitchell and Ruess [9].

**Post-fire N-balance.** There are several mechanisms for N-loss and N-gain in the boreal forest ecosystem. To determine the effect of post-fire alder N-input on total ecosystem N-balance, we estimated post-fire N-loss by multiplying rates of wildfire N-volatilization [17,39] by the amount of combustible N [17,40]. We then estimated N-gains using values from previous studies and this study. N-gains included the dominant sources of biological N-fixation (BNF) input in the boreal forest: feathermoss N-fixation input [41,42], free-living soil bacteria N-fixation input [43], *Peltigera* spp. N-fixation input [44], and Siberian alder N-fixation input from this study. Feathermoss N-fixation input was calculated by multiplying feathermoss cover values from this study by feathermoss N-fixation rates from other studies in the boreal forest [41,42]. Estimates of N-deposition [45] were also added to total N-gains. We then calculated within-stand post-fire N-balance by computing the difference between estimates of N-loss during fire and N-gain during the first 138-yr fire return interval (FRI). The average FRI for areas below 800 m on the Yukon-Tanana Uplands is 138 years [46]. It is important to note that our aim was to calculate N-exchange between terrestrial and atmospheric pools of N rather than estimating N-turnover within the terrestrial system that may include processes ultimately affecting N retention, such as nitrification.

## Statistical analysis

**Non-parametric density estimator.** Alders in this study were non-randomly aggregated across the landscape; therefore, alder density was determined using distance to the nearest alder at each plot [WDF (n = 80 plots) and the BF (n = 183 plots)] and a nonparametric estimator of density [28–30] implemented with the R function *np.density.est* [30]. A nonparametric median test was used to identify significant differences in nearest alder distance by post-fire age (11 vs 44 years) and among stand types within each burn scar.

**Scaling-up nodule-level N-fixation.** Alder annual plant-level N-fixation input (g N m$^{-2}$ plant$^{-1}$ yr$^{-1}$) was calculated by combining nodule-level N-fixation rates with plant-level live nodule biomass. The method for calculating plant-level N-fixation input follows established methods which account for seasonal variation in nodule-level N-fixation rates [4,31,32] and the assumption of 24-hour N-fixation that has been documented in other studies [47,48].

Annual stand-level N-fixation input (also referred to as alder N-fixation input) (kg N ha$^{-1}$ yr$^{-1}$) was calculated by multiplying the average area of nodulation for a plant within a stand

type ($m^2$ plant$^{-1}$) by alder annual plant-level N-fixation input and stand type alder density, as determined with the 2015 plot data [WDF (n = 21 plots) and the BF (n = 19 plots)].

**Alder N-input and growth traits by burn scar and stand type.** The ANOVA and Kruskal-Wallis tests were used to assess for significant differences in alder density, nodule-level N-fixation rate, alder growth traits (nodule biomass, ramet height, ramet diameter, specific leaf mass, number of live ramets, and number of dead ramets), and plant-level N-fixation input by burn scar and stand type. Unless stated otherwise, all statistical tests were conducted in R [49] and statistical significance was determined at α = 0.05. The Shapiro-Wilk test was used to assess the normality of variables, and non-normal variables were transformed using the Box-Cox power transformation. In the BF, non-normal variables included nodule-level N-fixation rate, live nodule biomass, dead nodule biomass, mean ramet diameter, mean ramet height, depth of the organic horizon, depth of the fibric layer, depth of the sapric layer, soil N, soil C, soil P, and plant-level N-fixation input. In the WDF non-normal variables included depth of the hemic layer, depth of the organic horizon, soil N, soil C, soil P, soil temperature, soil moisture, soil pH, mean ramet diameter, mean ramet height, dead nodule biomass, specific leaf mass, plant-level N-fixation input, and elevation. At the regional scale (both burn scars), non-normal variables included live nodule biomass, dead nodule biomass, mean ramet diameter, mean ramet height, live ramets per plant, and dead ramets per plant, soil moisture, depth of organic horizon, soil pH, depth of hemic layer, and solar radiation.

Homogeneity of variance across factor levels of categorical variables was tested with Levene's test. One-way ANOVAs were used to test for differences in alder N-input and growth traits between burn scars and among stand types. For significant ANOVA results, Tukey's test of honest significant difference was used to test factor level differences. For non-normal variables that could not be adequately transformed, differences were tested with the Kruskal-Wallis test and the Dunn-Bonferroni post-hoc test. Descriptive statistics throughout the text are untransformed and expressed as the mean ± 1 standard error, except alder density (plants ha$^{-1}$) and alder annual stand-level N-fixation inputs (kg N ha$^{-1}$ yr$^{-1}$), which are both reported as the mean ± 1 standard deviation. The alder growth traits were intercorrelated (all $|r| > 0.3$, $p < 0.05$). We therefore used principal component analysis (PCA) of alder growth traits to distil suites of correlated variables into one or few variables. An integrator growth variable was created through PCA in SPSS [50] using the 2015 plots [WDF (n = 21 plots) and the BF (n = 19 plots)] as sampling units and we then used this integrator variable to observe the relationship among alder growth traits, burn scars, vegetation stand types, and environmental characteristics [50]. The KMO Measure of Sampling Adequacy and Bartlett's test of sphericity were used to test each variable's sample size and if the matrix was an identity matrix.

**Multiple linear regression and AICc model selection.** Modeling was used to determine the significance and relative importance of predictors of alder density (as represented by distance to nearest alder), nodule-level N-fixation rate, alder growth PCA axes, plant-level live nodule biomass, and annual plant-level N-fixation input. Response variables were modeled separately in R using the 2015 plot data [WDF (n = 21 plots) and the BF (n = 19 plots)], multiple linear regression, and Akaike information criterion (AICc) best model subset multi-model inference. The response variables were modeled across both burn scars and within each burn scar using environmental characteristics, post-fire age (regional models only), and fire severity (dNBR) (BF models only) as predictors. Potential predictors included plot-level environmental variables, post-fire age (only for models that include both burn scars), and fire severity (BF models only). The potential predictors were normalized, tested for significant correlation with the response (Pearson's, $p < 0.05$), and non-collinear predictors ($|r| < 0.6$) were included in a global model. Post-fire age was included in all regional-scale global models (i.e., models that include both burn scars), and fire severity (dNBR) was included in all BF global models.

Separate global models were created for each response variable at the regional-scale (*n* = 6 global models) and for each burn scar (*n* = 12 global models). Best model subsets (AICc ≤ 2 units of lowest AICc) were selected with the *dredge* function from the *MuMIn* package in R [51], and β coefficients were standardized following Cade [52] and then averaged with the *model.avg* function from the *MuMIn* package.

**Structural equation models.** Multiple linear regression and AICc model selection did not indicate a direct effect of fire severity on any of the response variables. Therefore, we used structural equation models (SEM) to detect direct and indirect effects of fire severity on alder density, growth traits (PCA1 and PCA2), and N-fixation in the black spruce plots of the BF (n = 11). We created SEMs only for the black spruce stands because our landscape-level sample design captured the entire gradient of fire severity only in this stand type.

SEM predictors included significant predictors from the AICc best model subsets and fire severity. SEM models were fit using the *lavaan* package in R [53]. Non-significant ($p > 0.05$) variables in the SEM models were sequentially removed until only significant predictors remained. Modification indices were used to identify ecologically significant missing paths [54] that were not initially included in the AICc best model subset. SEM model fitness was determined using the chi-square test ($p > 0.05$), the root mean square error of approximation (RMSEA; lower 90% confidence intervals of RMSE close to zero), and the comparative fit index (CFI > 0.9) [54].

## Results

### Stand type classification

Hierarchical clustering and indicator species analysis of the relevé plot data produced three distinct post-fire stand types for the WDF that were named according to their dominant tree species: Black Spruce, Deciduous, and Mixed (black spruce and deciduous codominance) (Table 1). In the BF, four post-fire stand types emerged and were named according to their dominant pre-fire stand type and level of fire severity (Deciduous-Moderate, Black Spruce-Moderate, Black Spruce-Moderate to High, Black Spruce-High) (Table 1).

### Siberian alder density

Alder density was low in severely burned stands during early post-fire succession, but it likely increases by intermediate post-fire succession. Alder density in the younger BF (65 ± 14 plants ha$^{-1}$) was 150% lower than the WDF (162 ± 39 plants ha$^{-1}$) ($p < 0.001$, Fig 1). Between-fire differences were driven by very low alder density in the Black Spruce-High stand type within the BF (2 ± 1 plant ha$^{-1}$) (Fig 1). Alder density in the Black Spruce-High stand type was roughly 97% lower compared to Black Spruce-Moderate ($p = 0.013$) and Black Spruce-Moderate to High ($p < 0.001$) stand types (59 ± 27 and 61 ± 20 plants ha$^{-1}$, respectively), and 99% lower than the Deciduous-Moderate stand type (164 ± 44 plants $^{-1}$) ($p < 0.001$, Fig 1). Within the WDF, alder density in the Black Spruce stand type (98 ± 34 plants ha$^{-1}$) was 34% and 50% lower compared to the Deciduous and Mixed stand types (154 ± 53 and 195 ± 74 plants ha$^{-1}$, respectively), though differences were not significant ($p = 0.478$, Fig 1).

Regional-scale alder density was associated with an interaction between fire severity and environmental characteristics 11 years after fire. A strong effect of site-specific environmental characteristics on alder density was present after 40 years of post-fire succession. Across both burn scars, plots with high moisture in the organic horizon were associated with lower alder density (β = -6.68), $p = 0.012$) (Table 2). Within the BF, alder density was highest in plots that had a deeper Oe layer (β = 7.24), $p = 0.048$) (Table 2). A SEM for black spruce dominant plots showed a significant ($p = 0.026$) indirect negative effect of fire severity on alder density via O

**Table 1. Wickersham dome fire and the boundary fire stand types.**

| Stand type (% of burn scar study area) | dNBR | Indicator species or other cover type |
|---|---|---|
| **WICKERSHAM DOME FIRE** | | |
| Black Spruce (42%) | NA | Moss, *Rhododendron groenlandicum*, *Vaccinium vitis-idaea*, *Picea mariana* (seedling), *Betula glandulosa*, *Sphagnum* spp., *Equisetum* spp., *Eriophorum* spp., *Rubus chamaemorus*, *Polygonum alpinum*, *Rhododendron palustre* ssp. *decumbens* |
| Deciduous (39%) | NA | Litter, *Betula neoalaskana* (tree), *Populus tremuloides* (tree), dead and down trees, *Populus tremuloides* (seedling), *Rosa acicularis*, *Geocaulon lividum*, *Picea glauca* (tree) |
| Mixed (19%) | NA | Lichen, *Picea mariana* (tree), *Vaccinium uliginosum*, *Salix* spp., *Cornus canadensis*, *Empetrum nigrum* |
| **BOUNDARY FIRE** | | |
| Deciduous-Moderate (28%) | 354 ± 30 [a] | *Betula neoalaskana* (tree), *Betula neoalaskana* (seedling), litter, *Calamagrostis* spp., dead and down trees, *Populus tremuloides* (seedling), *Chamerion angustifolium*, *Rubus idaeus*, *Cornus canadensis*, *Populus tremuloides* (tree), *Rosa acicularis*, *Lycopodium* spp. |
| Black Spruce-Moderate (32%) | 350 ± 35 [a] | *Sphagnum* spp., *Rubus chamaemorus*, *Picea mariana* (tree), Lichen, *Rhododendron palustre* ssp. *decumbens*, *Eriophorum* spp., *Vaccinium vitis-idaea*, *Picea mariana* (seedling), *Vaccinium oxycoccos*, Moss, *Betula nana*, *Empetrum nigrum*, *Andromeda polifolia*, *Polygonum alpinum*, *Petasites frigidus* |
| Black Spruce-Moderate to High (20%) | 499 ± 28 [d] | *Vaccinium uliginosum*, *Rhododendron groenlandicum*, *Betula glandulosa*, *Betula* sp. (hybrid shrub), *Arctagrostis latifolia* |
| Black Spruce-High (20%) | 664 ± 28 [c] | *Salix* spp. and *Carex* spp. |

dNBR classes for the Boundary Fire are low (25 to 275), moderate (276 to 549), and high ($\geq$ 550) and values represent the mean ± standard error. Different letters among dNBR values indicate significant differences at $p < 0.05$. Indicator species are listed in order of descending indicator value.

soil depth (Fig 2A), where higher severity fire results in more combustion of the organic layer. Within the WDF, variation of alder density across all plots was negatively associated with moisture in the organic horizon ($\beta$ = -9.83) and positively associated with slope ($\beta$ = 7.30) ($p$ = 0.005 and $p$ = 0.034, respectively); wet and/or shallow-sloped plots had lower alder density than drier and/or steeper-sloped plots (Table 2).

## Alder growth traits and plant-level N-fixation input

Alder individuals fixed lower amounts of N per year in the intermediate burn scar compared to the early burn scar, especially within black spruce dominant stands—likely due to lower rates of nodule-level N-fixation rather than lower nodule biomass. Mean nodule-level N-fixation rate in the BF (12.88 ± 1.18 $\mu$mol N g$^{-1}$ hr$^{-1}$) was 70% higher than in the WDF (7.58 ± 0.59 $\mu$mol N g$^{-1}$ hr$^{-1}$) ($p < 0.001$), (Fig 1). Within the BF, nodule-level N-fixation rate was not significantly different among stand types ($p > 0.05$) (Fig 1). Within the WDF, nodule-level N-fixation rate in Black Spruce stands (5.85 ± 1.04 $\mu$mol N g$^{-1}$ hr$^{-1}$) was not significantly different than Deciduous stands ($p$ = 0.433), but was 37% lower than Mixed stands ($p$ = 0.020) (7.52 ± 0.43 and 9.34 ± 0.77 $\mu$mol N g$^{-1}$ hr$^{-1}$, respectively) (Fig 1). We did not detect a significant difference ($p$ = 0.384) in plant-level live nodule biomass between the BF (16.30 ± 3.56 g nodule m$^{-2}$ plant$^{-1}$) and the WDF (9.45 ± 1.58 g nodule m$^{-2}$ plant$^{-1}$), nor among stand types

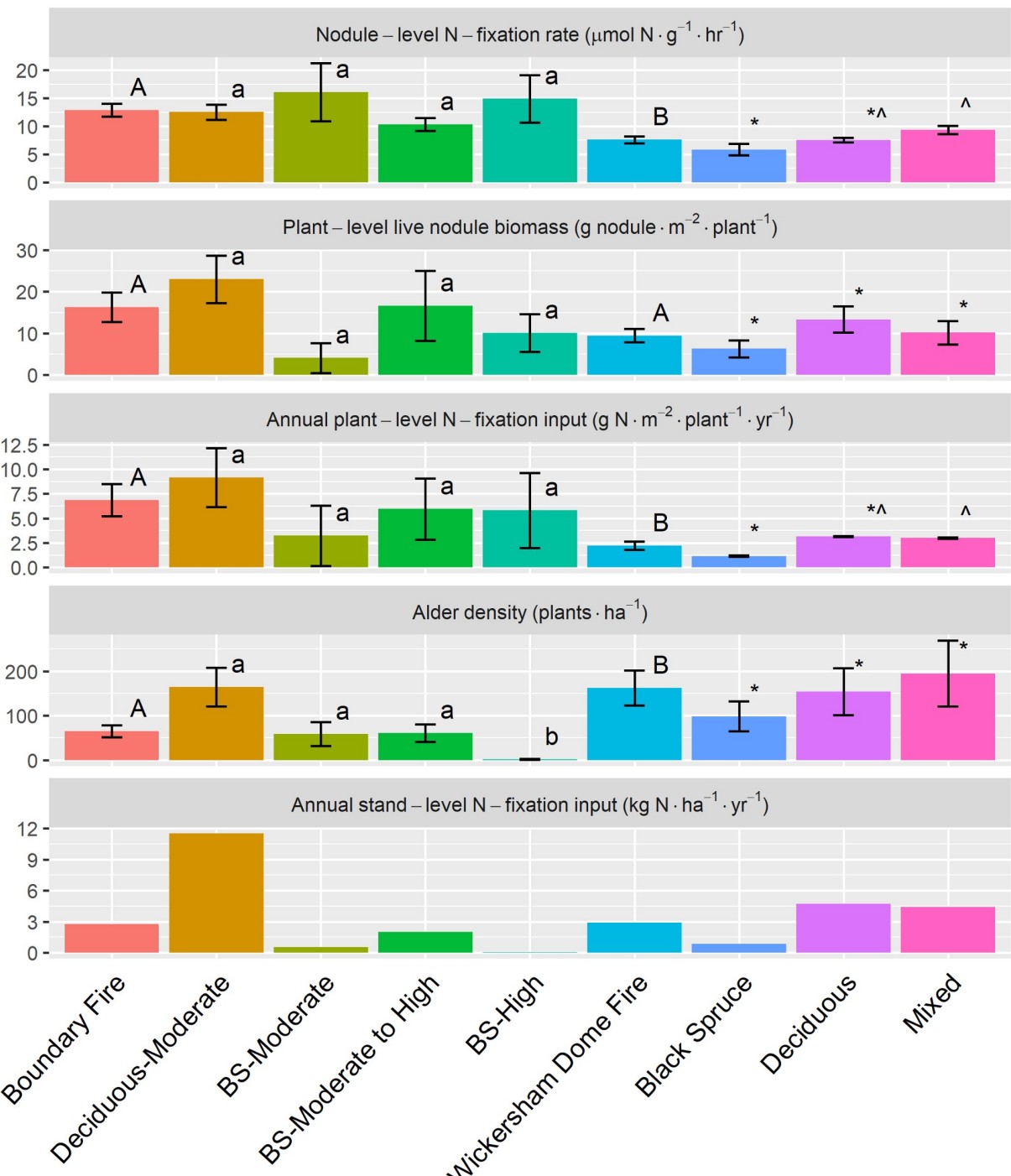

**Fig 1. Siberian alder nodule productivity, plant-level N-input, plant density, and stand-level N-fixation input in the Boundary Fire (n = 19), Wickersham Dome Fire (n = 21), and stand types within each burn scar.** Significant differences are determined at $p < 0.05$ between burn scars (A or B), and among stand types of the Boundary Fire (a or b) and Wickersham Dome Fire (* or ^). Graph order from top to bottom: nodule-level N-fixation rate ($\mu$mol N g$^{-1}$ hr$^{-1}$), plant-level live nodule biomass (g nodule m$^{-2}$ plant$^{-1}$), annual plant-level N-fixation input (g N m$^{-2}$ plant$^{-1}$ yr$^{-1}$), alder density (plants ha$^{-1}$), and annual stand-level N-fixation input (kg N ha$^{-1}$ yr$^{-1}$).

**Table 2. Landscape models (AICc) for Siberian alder N-input and growth traits.**

| Response | Predictor Variable (standardized beta coefficient, importance value) |
| --- | --- |
| **REGION** | |
| Siberian alder density | **Moisture in organic horizon (-6.68, 1)**, mineral horizon C:N (-3.05, 0.38) |
| PCA1 | **Depth of organic horizon (-0.64, 1), depth of hemic layer (0.29, 1)**, *mineral horizon N:P ratio (-0.21, 0.52)*, post-fire age (0.19, 0.52), mineral horizon pH (0.14, 0.13) |
| PCA2 | **Post-fire age (0.43, 1), moisture in organic horizon (-0.28, 0.84)**, *mineral horizon C:N ratio (-0.27, 0.51)*, mineral horizon P (-0.19, 0.16), mineral horizon bulk density (0.18, 0.14) |
| Plant-level live nodule biomass | **Depth of organic horizon (-7.08, 1)**, post-fire age (-2.27, 0.46) |
| Nodule-level N-fixation rate | **Post-fire age (-2.68, 1)** |
| Annual plant-level N-fixation input | **Depth of organic horizon (-2.16, 1), post-fire age (-2.02, 1)**, soil N:P ratio (-1.24, 0.56), depth of hemic layer (-0.89, 0.37) |
| **BOUNDARY FIRE** | |
| Siberian alder density | **Depth of hemic layer (7.24, 0.77)**, *depth of sapric layer (-6.49, 0.50), dNBR (-6.32, 0.47)* |
| PCA1 | **Depth of organic horizon (-0.42, 0.99), mineral horizon pH (0.34, 0.95), moisture in organic horizon (-0.34, 0.82)** |
| PCA2 | **Elevation (0.34, 0.99), mineral horizon pH (0.29, 0.96)** |
| Plant-level live nodule biomass | **Depth of organic horizon (-8.08, 0.82), mineral horizon N:P (-6.41, 0.61)**, *moisture in organic horizon (-5.69, 0.33), mineral horizon pH (5.05, 0.15)*, slope (-4.74, 0.12) |
| Nodule-level N-fixation rate | *Mineral horizon C:N ratio (-2.37, 0.52)*, slope (-1.98, 0.47), dNBR (1.76, 0.34), mineral horizon bulk density (1.82, 0.18) |
| Annual plant-level N-fixation input | **Mineral horizon N:P ratio (-3.55, 0.78), depth of organic horizon (-3.08, 0.78)**, *slope (-2.77, 0.22)* |
| **WICKERSHAM DOME FIRE** | |
| Siberian alder density | **Moisture in organic horizon (-9.83, 0.94), slope (7.30, 0.74)** |
| PCA1 | **Depth of organic horizon (-0.59, 0.94)** |
| PCA2 | **Mineral horizon C:N ratio (-0.58, 1)**, moisture in organic horizon (-0.34, 0.5) |
| Plant-level live nodule biomass | NA |
| Nodule-level N-fixation rate | **Elevation (1.90, 1)** |
| Annual plant-level N-fixation input | *Elevation (0.77, 0.69)*, Depth of organic horizon (-0.67, 0.54) |

Response variables: Siberian alder density (plants ha$^{-1}$), PCA1 [plant-level live nodule biomass (+), mean ramet height (+), mean ramet diameter (+), and specific leaf mass (-)], PCA2 [number of live ramets per plant (+) and dead ramets per plant (+)], plant-level live nodule biomass (g nodule m$^{-2}$ plant$^{-1}$), nodule-level N-fixation rate (μmol N g$^{-1}$ hr$^{-1}$), and annual plant-level N-fixation input (g N m$^{-2}$ plant$^{-1}$ yr$^{-1}$) across the region (n = 40), the Boundary Fire (n = 19), and the Wickersham Dome Fire (n = 21). Standardized beta coefficients and importance values are in parentheses, respectively, for each predictor variable. The baseline level for post-fire age in the regional models is the younger Boundary Fire. Significant predictors (p < 0.05) are in bold. Marginally significant predictors (p < 0.1) are italicized.

within the BF or WDF (p = 0.230 and p = 0.225, respectively, Fig 1). However, alder annual plant-level N-fixation input in the BF (6.86 ± 1.65 g N m$^{-2}$ plant$^{-1}$ yr$^{-1}$) was significantly higher than the WDF (2.26 ± 0.03 g N m$^{-2}$ plant$^{-1}$ yr$^{-1}$) (p = 0.036, Fig 1). Within the WDF, alder annual plant-level N-fixation input varied by stand type—Black Spruce stands (1.17 ± 0.07 g N m$^{-2}$ plant$^{-1}$ yr$^{-1}$) were roughly 60% lower than Deciduous and Mixed stands (3.17 ± 0.04 and 3.00 ± 0.07 g N m$^{-2}$ plant$^{-1}$ yr$^{-1}$, respectively); however the difference between Black Spruce and Mixed stands was significant whereas the Black Spruce and Deciduous stand difference

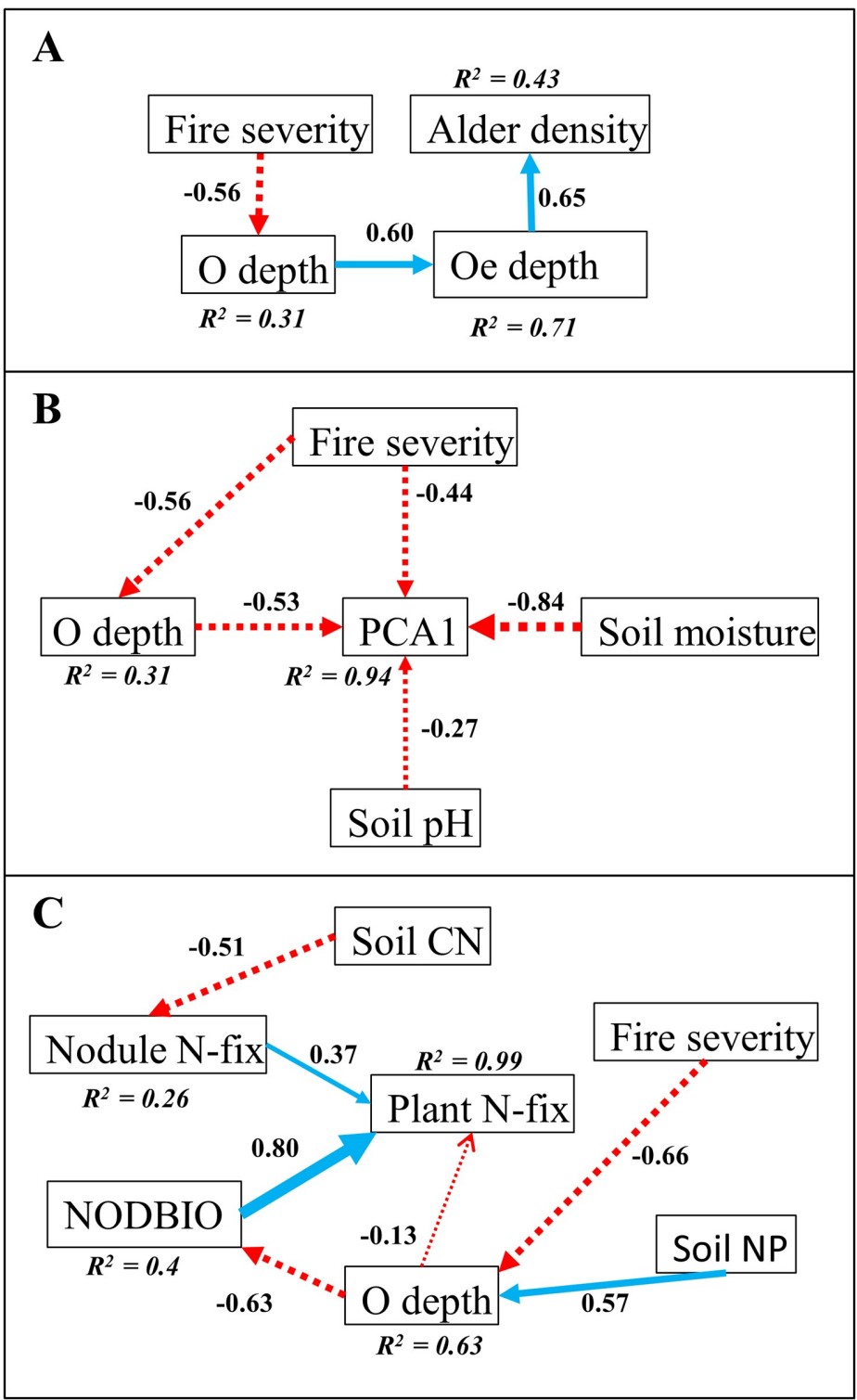

**Fig 2. Structural equation model of A) Siberian alder density (plants ha$^{-1}$), B) PCA1 [plant-level live nodule biomass (+), mean ramet height (+), mean ramet diameter (+), and specific leaf mass (-)], and C) Plant N-fix [annual plant-level N-fixation input (g N m$^{-2}$ plant$^{-1}$ yr$^{-1}$)] in post-fire black spruce dominant plots of the Boundary Fire (n = 11).** Fire severity = difference in normalized burn ratio (dNBR), O depth = depth of organic horizon, Oe depth = depth of hemic layer, Soil pH = mineral horizon pH, Soil moisture = moisture in organic horizon, Soil CN = mineral horizon C:N ratio, Nodule N-fix = nodule-level N-fixation rate (μmol N g nodule$^{-1}$ hr$^{-1}$),

NODBIO = plant-level live nodule biomass (g nodule m$^{-2}$ plant$^{-1}$), Soil NP = mineral horizon N:P ratio. Standardized beta coefficients are shown for predictor variable pathways. Negative pathways are shown in red dotted lines, and positive pathways are shown in blue solid lines. Wider lines indicate stronger beta coefficients.

was not significant (p = 0.043 and p = 0.082, respectively, Fig 1). The low sample size within Deciduous stands (n = 5) may explain the lack of significant difference from Black Spruce stands (S1 Table). We did not detect significant differences in alder annual plant-level N-fixation input among stand types of the BF (p = 0.395, Fig 1).

Higher alder growth in deciduous dominated versus black spruce dominated stands reflects general patterns of boreal forest aboveground annual primary productivity. Additionally, the height, diameter, and number of alder ramets per plant was strongly influenced by time since fire as opposed to environmental characteristics. The PCA of alder growth traits resulted in two significant axes that explained 48% and 26% of alder growth, respectively (S2 Table). Alder plant-level live nodule biomass, mean ramet height, and mean ramet diameter loaded positively onto the first PCA axis (hereafter referred to as *PCA1*), whereas specific leaf mass loaded negatively onto PCA1. The number of live ramets per plant and dead ramets per plant both loaded positively onto the second PCA axis (hereafter referred to as *PCA2)*. Dead nodule biomass was excluded from the PCA due to insufficient sample size (KMO < 0.5).

Higher values of PCA1 indicate plots with alders that were relatively tall, with greater basal diameter, thinner leaves, and higher plant-level live nodule biomass (Fig 3). Across the region, the BF and WDF did not have significantly different PCA1 values (p = 0.708, Fig 3A) (S3 Table). Significantly higher PCA1 values were observed in Deciduous-Moderate stands compared to Black Spruce-Moderate stands in the BF (p = 0.026, Fig 3B) (S4 Table), and in Deciduous versus Black Spruce stands of the WDF (p < 0.001, Fig 3B) (S1 Table).

The number of live ramets per plant and dead ramets per plant both loaded positively onto the second PCA axis (hereafter referred to as *PCA2*). PCA2 was significantly higher in the WDF than in the BF (p = 0.001, Fig 3A, S3 Table), which reflects alder's characteristic sprouting of new ramets over time. PCA2 did not vary significantly among stand types within the BF or among stand types within the WDF (p = 0.498 and p = 0.642, respectively) (Fig 3B, S4 and S1 Tables).

Eleven years after fire (BF), alders growing in a shallow, low-moisture organic horizon with higher pH of the mineral horizon had more live nodules, more ramets, and larger ramets compared to other alders. However, alder growth in the BF was lower on black spruce dominant plots that experienced high severity fire—suggesting that high severity fire had a stronger effect on alder density than soil conditions. Forty years after fire (WDF), alder showed higher growth (height, diameter, and more numerous ramets and nodules) in shallow organic horizons, especially where mineral horizon C:N ratios were low, such as deciduous- or mixed-dominated stands of higher aboveground annual primary productivity. Yet, the level to which alder may drive soil conditions rather than respond to them is unclear.

Across both burn scars, PCA1 [a composite variable of alder plant-level live nodule biomass (+), mean ramet height (+), mean ramet diameter (+) and specific leaf mass (-)] was highest in plots of low depth of the organic horizon (β = -0.64, p < 0.001) and high depth of the hemic layer (β = 0.29, p = 0.017) (Table 2, Fig 3A). Within the BF, higher mineral horizon pH (β = 0.34, p = 0.026) and low moisture in the organic horizon (β = -0.34, p = 0.033) were additional predictors of high PCA1 (Table 2). SEM models for black spruce dominant plots in the BF (n = 11) indicated a direct negative effect of fire severity on PCA1 (p < 0.001), but a net positive effect of fire severity on PCA1 through its negative effects on O soil depth (p = 0.026) (Fig 2B). In contrast, PCA1 in the WDF was solely associated with O soil depth (β = -0.59)

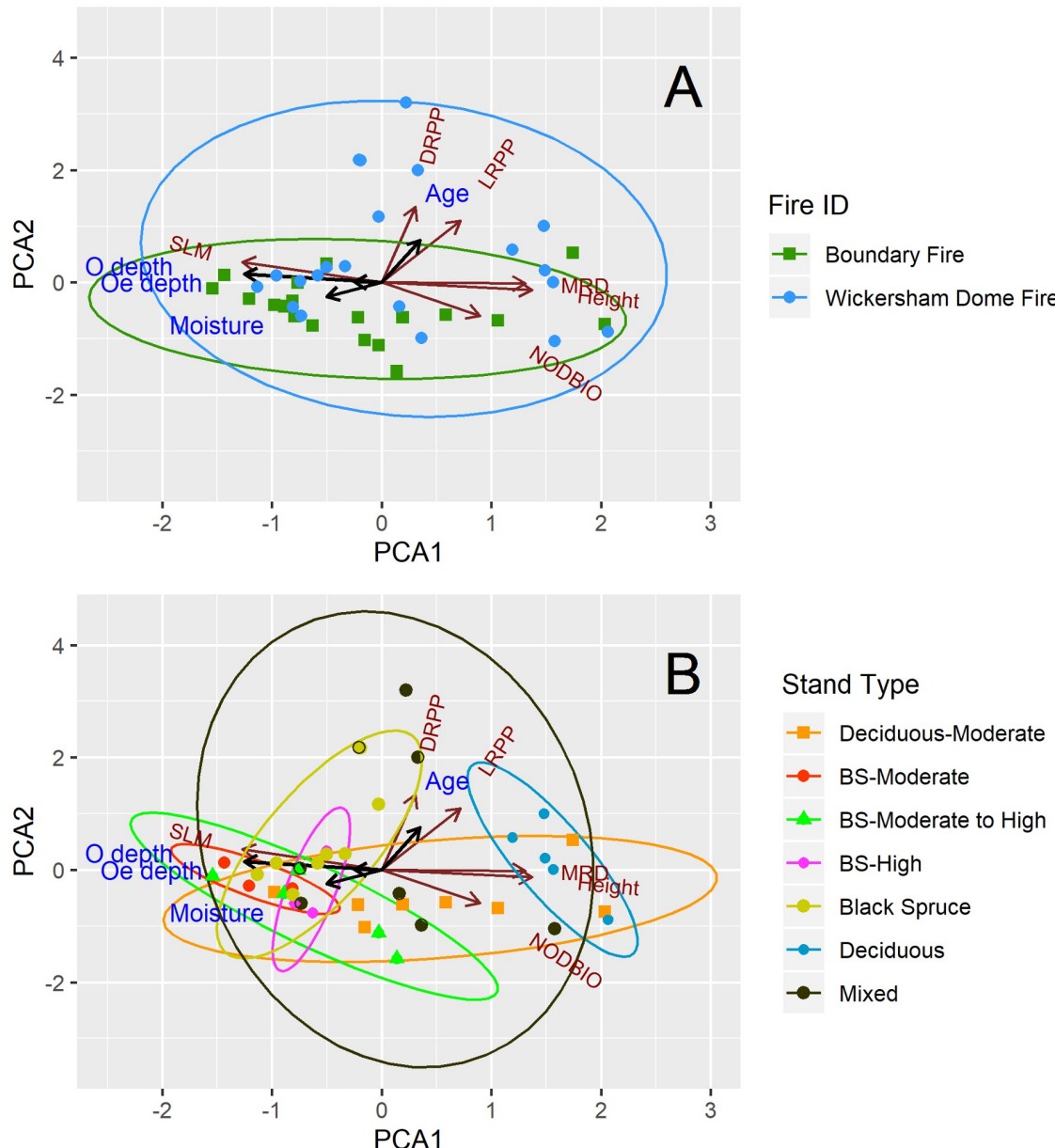

**Fig 3. Principal component analysis (PCA) of Siberian alder growth traits: Plant-level live nodule biomass (NODBIO), mean ramet height (Height), mean ramet diameter (MRD), specific leaf mass (SLM), live ramets per plant (LRPP), and dead ramets per plant (DRPP).** The PCA is displayed by burn scar (A) and by stand type within each burn scar (B). Brown arrows and labels represent the orientation of Siberian alder growth traits used in the PCA. Ellipses represent the 95% confidence interval for each burn scar or stand type. Black arrows with associated blue labels symbolize the important predictor variables of each PCA axis as determined by AICc best model subset for both burn scars: Moisture = moisture in organic horizon, O depth = depth of organic horizon, Oe depth = depth of hemic layer, Age = post-fire age.

(p = 0.007) (Table 2). Though post-fire age had no effect on PCA1 regionally (p = 0.120) (Table 2), differences in which variables predicted PCA1 across both burn scars suggest that environmental variables promote early post-fire growth although those variables are not as important to alder growth by intermediate succession.

PCA2 [a composite of live ramets (+) and dead ramets (+)] was strongly associated with post-fire age (β = 0.43, p = 0.003) and moisture in the organic horizon (β = -0.28, p = 0.048)

across both burn scars (p < 0.05, Table 2, Fig 3A). Within the younger BF, higher elevation (β = 0.34) and higher soil pH (β = 0.29) were both positively associated with higher PCA2 (p = 0.001 and p = 0.005, respectively) (Table 2). Within the WDF, however, PCA2 was best predicted by an inverse relationship with mineral horizon C:N ratio (β = -0.58, p = 0.011) (Table 2).

## Alder annual stand-level N-fixation input

Fire appears to alter site conditions to favor higher alder annual plant-level N-fixation input within all stand types, but it also reduced alder density and therefore lowered alder annual stand-level N-fixation input—especially within severely burned black spruce dominated stands. Estimates of alder annual stand-level fixation inputs are a product of alder annual plant-level N-fixation input, alder density, and area of nodulation. As a result of higher alder annual plant-level N-fixation input in the BF compared to the WDF (p = 0.036) and higher alder density in the WDF compared to the BF (p < 0.001) there was no significant difference in alder annual stand-level N-fixation inputs between these two burn scars (2.75 ± 0.08 and 2.91 ± 0.06 kg N ha$^{-1}$ yr$^{-1}$, respectively) (Fig 1). However, within the WDF, alder annual stand-level N-fixation input in the Black Spruce stand type (0.83 ± 0.04 kg N ha$^{-1}$ yr$^{-1}$) was 82% lower compared to the Deciduous stand type (4.72 ± 0.02 kg N ha$^{-1}$ yr$^{-1}$) (Fig 1). Within black spruce dominant stands of the BF, alder annual stand-level N-fixation inputs averaged 93% lower (0.86 ± 0.03 kg N ha$^{-1}$ yr$^{-1}$) than the Deciduous stand type (11.53 ± 0.22 kg N ha$^{-1}$ yr$^{-1}$) (Fig 1). In the BF, alder annual stand-level N-fixation input within the Black Spruce-High stand type (0.06 ± 0.003 kg N ha$^{-1}$ yr$^{-1}$) averaged 88% and 97% lower than Black Spruce-Moderate and Black Spruce-Moderate to High stand types, respectively (Fig 1).

Environmental characteristics associated with alder annual plant-level N-fixation input interact across spatiotemporal scales. Local plot conditions (e.g. soil chemistry and O soil depth) were influenced by fire severity and were associated with alder annual plant-level N-fixation input eleven years after fire. Forty years after fire broader-scale landscape variables (e.g. topography) were associated with alder annual plant-level N-fixation input. Across both burn scars, depth of organic horizon (β = -2.16) and post-fire age (β = -2.02) were the strongest predictors of alder annual plant-level N-fixation input (p < 0.011 and p = 0.008, respectively) (Table 2). The components of alder annual plant-level N-fixation input (plant-level live nodule biomass and nodule-level N-fixation rate) were separately associated with depth of organic horizon (β = -7.08, p < 0.001) and post-fire age (β = -2.68, p < 0.001), respectively, across both burn scars (Table 2). Within the younger BF, alders growing in a low mineral horizon N:P ratio (β = -3.55, p = 0.019) and shallow organic horizon (β = -3.08, p = 0.034) had the highest alder annual plant-level N-fixation input—a result largely due to the significantly higher plant-level live nodule biomass in shallow organic horizons (β = -8.08, p = 0.022) and low mineral horizon N:P ratios (β = -6.41, p = 0.035) (Table 2). A SEM of alder annual plant-level N-fixation input for black spruce dominant plots in the BF (n = 11) showed that fire severity had a direct negative effect on depth of organic horizon (p < 0.001) and therefore an indirect positive effect on live nodule biomass (Fig 2C). Thus, high fire severity had a positive, indirect association with alder annual plant-level N-fixation input in the BF (Fig 2C). Within the WDF, alders at lower elevations (valley bottoms) had lower nodule-level N-fixation rates compared to alders at higher elevations (ridgetops) (β = 1.90, p < 0.001), resulting in a marginal effect on annual plant-level N-fixation input (β = 0.77, p = 0.077) (Table 2).

**Table 3. Estimates of post-fire N-balance by stand type and fire severity.** Feathermoss N-fixation input (kg N ha$^{-1}$ yr$^{-1}$), free-living soil bacteria N-fixation input (kg N ha$^{-1}$ yr$^{-1}$), *Peltigera* ssp. N-fixation input (kg N ha$^{-1}$ yr$^{-1}$), Siberian alder N-fixation input (kg N ha$^{-1}$ yr$^{-1}$), and N-deposition (kg N ha$^{-1}$ yr$^{-1}$) are shown by stand type and age [early (0–20 yr), mid (20–60 yr), and late (60–138 yr)]. The total of all N-inputs during a 138 yr fire return interval (FRI) (kg N ha$^{-1}$) is shown by stand type. N-loss due to volatilization (kg N ha$^{-1}$) and N-balance (kg N ha$^{-1}$) are shown by stand type and fire severity.

| Stand Type and Age | Feathermoss N-fixation input[1] | Free-living soil bacteria N-fixation input[2] | *Peltigera* spp. N-fixation input[3] | Siberian alder N-fixation input | N-dep.[5] | FRI N-input[6] | Low Severity | | Moderate Severity | | High Severity | |
|---|---|---|---|---|---|---|---|---|---|---|---|---|
| | | | | | | | N-loss[7] | N-balance[8] | N-loss[7] | N-balance[8] | N-loss[7] | N-balance[8] |
| **Black Spruce (self-replacing)** | | | | | | | | | | | | |
| Early | 0.27–1.67 | 0.84–0.99 | 0.22–0.94 | 0.52 ± 0.03 | 0.3 | 486 | 270 | 216 | 900 | -414 | NA | NA |
| Mid | 0.68–4.25 | | | 0.83 ± 0.04 | | | | | | | | |
| Late | 1.42–7.66 | | | 0.83 ± 0.04[4] | | | | | | | | |
| **Black Spruce to Deciduous (conversion)** | | | | | | | | | | | | |
| Early | 0–0.15 | 0.95–1.44 | 0.02–0.12 | 0.06 ± 0.003 | 0.3 | 195 | NA | NA | NA | NA | 1350 | -1155 |
| Mid | 0.01–0.48 | | | 0.06 ± 0.003[4] | | | | | | | | |
| Late | 0.03–0.54 | | | 0.06 ± 0.003[4] | | | | | | | | |
| **Deciduous (self-replacing)** | | | | | | | | | | | | |
| Early | 0–0.15 | 0.95–1.44 | 0.02–0.12 | 11.53 ± 0.22 | 0.3 | 974 | 84 | 890 | NA | NA | 227 | 747 |
| Mid | 0.01–0.48 | | | 4.72 ± 0.02 | | | | | | | | |
| Late | 0.03–0.54 | | | 4.72 ± 0.02[4] | | | | | | | | |

[1] Estimated using stand-level feathermoss N-fixation rates from other studies [41,42] and feathermoss cover (%) in stands of this study

[2] Free-living soil bacteria N-fixation input for similar stand types in Nohrstedt [43]

[3] Estimated from rates of *Peltigera* spp. N-fixation input in similar stand types of Katmai National Park and Preserve [44]

[4] Direct measurement unavailable; estimates were made with N-fixation input values from younger stand types

[5] N-deposition estimates from the National Atmospheric Deposition Program [59]

[6] Sum of feathermoss N-fixation input, free-living soil bacteria N-fixation input, *Peltigera* spp. N-fixation input, Siberian alder N-fixation input, and N-deposition in each stand type over a 138 yr FRI.

[7] Black Spruce and Black Spruce-to-Deciduous values were calculated using estimates of N-loss (%) for low [39], moderate [17], and high severity fire [17] and total combustible N [17]. Deciduous stand values were calculated using hypothetical values for N-loss (Low Severity: 10% of foliage, 50% of litter, 10% of soil organic layer; High Severity: 100% of foliage, 100% of litter, 20% of soil organic layer) and measured values for total combustible N [40].

[8] Difference between 138 yr post-fire FRI N-input, and N-loss during the preceding fire event.

## Post-fire N-balance

A positive post-fire N-balance (N-gains minus N-loss) is estimated in low to moderate severity fires after the first FRI, whereas high severity black spruce stands—where N-fixer density is very low and N-volatilization quite high—likely exhibit a negative N-balance. For a black spruce dominant stand which self-replaces after low severity fire, N gains are estimated to offset volatilized N over a subsequent 138 yr (FRI)—mostly due to feathermoss N-input (Table 3). For a deciduous stand that self-replaces after either low or high severity fire, N-gains are estimated to exceed volatilized N by an order of magnitude during a subsequent 138 yr FRI—mostly due to high Siberian alder N-input (Table 3). For a black spruce dominant stand that converts to deciduous dominance after high severity fire, N-gains are estimated to offset 15% of volatized N during a subsequent 138 yr FRI—an imbalance due to relatively high N-volatilization and relatively low BNF input, including that from Siberian alder (Table 3). Given the particularly low alder density after high severity fire, it is unlikely that our underestimation of plant-level live nodule biomass accounts for the large difference between N-loss and N-gain in high severity stands.

Other potential sources of post-fire N-gains may include 1–2 kg N ha$^{-1}$ yr$^{-1}$ from *Lupine* spp., 0.78 kg N ha$^{-1}$ yr$^{-1}$ from *Shepherdia canadensis*, or 9 kg N ha$^{-1}$ yr$^{-1}$ from *Myrica gale*

[55–57]; however, all three species occurred at either zero or trace percent cover in stands of this study. Thawed permafrost also presents another potential source of post-fire N-gains with nearly 30 kg N ha$^{-2}$ made available to vegetation from 1 cm of thawed permafrost [58]. Yet, less than 10% of plant-available N within thawing permafrost is taken-up by deep-rooted plants during late growing season [59].

## Discussion

We sought to disentangle the effects of fire severity from other potential effects on alder density (and therefore stand-level N-fixation input). We characterized how alder density, nodule-level N-fixation, nodule-level biomass, alder ramet and leaf traits, and plant-level N-fixation input vary across a fire severity gradient, fire age, and environmental characteristics (soil and topography). The results of this study support our hypothesis that high severity fires limit post-fire alder density and stand-level N-fixation inputs during secondary succession, but the association between fire severity and altered alder N-input is complex. We found that 11-years post-fire, high severity fire limits alder N-input via reductions to alder density; however, this effect was limited to black spruce dominant stands. Forty years after fire, alder annual stand-level N-fixation input in black spruce dominant stands was much lower than either deciduous or mixed stands largely due to lower nodule-level N-fixation rates that occurred in deep, wet organic horizons. Siberian alder annual stand-level N-fixation input varied among stand types and over time, with early-succession high severity black spruce dominant stands having the lowest inputs and early-succession moderately burned deciduous stands the highest inputs—a difference driven by variation in alder density and live nodule biomass along fire severity and soil chemistry gradients. Changing soil conditions coincide with differences in alder characteristics throughout secondary succession, making it difficult to parse alder's preferred soil conditions from alder's effect on soil conditions. However, the results from this study suggest that high severity fire and unfavorable soil conditions interact to limit post-fire alder N-input in the boreal forest. In converted black spruce stands these limitations on post-fire alder N-input, combined with high N-volatilization, result in net N-losses after the first fire return interval. However, potential alder recruitment and spread in these stands over successive fire return intervals may eventually recover N-losses from severe fire.

### Spatiotemporal variation in Siberian alder N-fixation input

Siberian alder annual stand-level N-fixation inputs (kg N ha$^{-1}$ yr$^{-1}$) in this study were comparable to values for Siberian alder growing in intermediate-age white spruce stands [9], but approximately half those reported for thin-leaf alder (*A. tenuifolia*) growing along intermediate-age boreal forest floodplains [4,33]. Yet, Siberian alder N-fixation input is comparable to thin-leaf alder N-fixation input at the patch-scale. Large, dense patches of Siberian alder (~ 1 ha) were periodically encountered in both the Deciduous-Moderate and Black Spruce-Moderate stand types of the BF, but none occurred in any of the N-fixation sampling plots. If we scale-up Siberian alder annual plant-level N-fixation inputs (g N m$^{-2}$ plant$^{-1}$ yr$^{-1}$) for these large patches they are estimated to fix approximately 91 ± 30 kg N yr$^{-1}$ and 33 ± 31 kg N yr$^{-1}$ for Deciduous-Moderate and Black Spruce-Moderate stand types, respectively. Therefore, post-fire Siberian alder is capable of fixing N at rates similar to thin-leaf alder of boreal forest floodplains [1,4,31,33].

Our estimate of stand-level N-fixation input is likely conservative due to the spatial restrictions we imposed on alder nodule sampling. Similar to previous studies, our measurements of plant-level live nodule biomass also were restricted to within a 1-meter radius of genets [9,31,33]. However, we and others [9] have observed nodules several meters from individual

plants in all stand types, and although at low density, these nodules were outside our sampling radius.

## Effect of fire severity on Siberian alder N-fixation input

Throughout boreal Alaska, fire severity, pre- and post-fire vegetation, depth of organic horizon, and environmental conditions are closely linked [60]. Our data suggest there is also a coupling between fire severity and alder N-input and growth traits in black spruce dominant stands after fire. We found that combustion of organic horizons during high severity fires was associated with a decrease in alder density, likely due to the destruction of rhizomes that are predominately found at the interface between the organic and mineral horizons [21,61]. Significantly lower alder density in severely burned black spruce dominant stands 11 years after the BF portends a link between the increasing incidence of high-severity fires and changing alder population dynamics across the Alaskan boreal forest.

Higher plant-level live nodule biomass and nodule-level N-fixation was observed in areas where fire destroyed the organic horizon. Lantz and others [20] showed that increased fire severity led to improved seedbeds and higher alder productivity. The accumulation of ash following the combustion of thick organic mats in black spruce dominant stands can increase available P in surface soils [62], and higher soil P is associated with increased nodule growth and higher annual plant-level N-fixation input by alder [4,31,33]. The Oi and Oe layers of the organic horizon were often combusted during the Boundary Fire, possibly contributing to increased soil aeration within the Oa layer—a condition that is known to increase nodule biomass [63] as well as N-fixation rates [33]. Despite our results which show fire can induce higher nodulation and nodule-level N-fixation, high severity fire in black spruce stands lowers alder density to an extent at which annual stand-level N-fixation inputs are considerably reduced. Though disturbances often lead to hotspots of alder seed recruitment and growth [20,64,65], a key question is whether the combination of lower alder density and higher alder nodulation and nodule-level N-fixation after severe fire will result in an overall increase or decrease of alder N-fixation inputs over longer time periods.

## Post-fire N-balance

Understanding factors influencing alder distribution, expansion and productivity are critical to forecasting post-fire N-balance in a changing boreal forest. Our N-balance estimates suggest that wildfire-induced stand conversions [15] resulting from a more severe fire regime [14] could be associated with declining landscape-level N pools in stands not underlain by permafrost. In stands that convert from black spruce to deciduous dominance, we estimated low BNF input—and therefore low N-gains—due to low density of alder and other N-fixers. High alder density, growth, and N-fixation have been documented within deciduous stands of this study and others [9,23,24], therefore alder is capable of high post-fire N-input in newly converted deciduous stands. Yet, alder populations not only need to establish in newly converted deciduous stands, but also persist over multiple fire cycles to offset the magnitude of N volatilized during stand conversion.

## Landscape-scale Siberian alder N-fixation input and ecological impacts

Our results highlight differences in alder annual stand-level N-fixation input among stand types, landscape positions, fire severity levels, and post-fire successional stands. The complex spatial distribution of factors influencing alder annual stand-level N-fixation input must be mapped at the regional scale to better assess post-fire N-balance across the landscape. By mapping just one of these factors (stand type), we found that 28% of the BF study area was

Deciduous-Moderate, 32% was Black Spruce-Moderate, 20% was Black Spruce-Moderate to High, and 20% was Black Spruce-High (Table 1). The distribution of stand types combined with their estimated N-balance throughout a single fire return interval suggests that a future fire regime of increased fire severity could lead to reconfigurations N across the landscape. Specifically, nearly one-third of the landscape (Deciduous-Moderate) would accumulate N due to very high alder density and productivity, while one-fifth (Black Spruce-High) would trend toward N-depletion without alder recruitment and spread. Progressive N-losses over several fire cycles could further expand areas trending toward N-depletion and significantly alter ecosystem patterns and processes within upland boreal forests. Future analyses of within-stand alder patch distribution as well as patch- and landscape-scale mapping of the factors affecting post-fire N-balance are needed to help predict ecosystem consequences of changing N-pools.

## Supporting information

**S1 Table. Descriptive statistics of the alder growth traits in the Wickersham Dome Fire.** Statistics were calculated using the 2015 dataset (n = 21). Variables significantly different (p < 0.05) across stand types are shown in bold print. Different letters in the same row indicate significant differences among stand types at p < 0.05. NODBIO = live nodule biomass (g nodule m$^{-2}$ plant$^{-1}$); Height = mean ramet height (m); SLM = specific leaf mass (mg cm$^{-2}$); MRD = mean ramet diameter (cm); LRPP = live ramets per plant; DRPP = dead ramets per plant; PCA1 = PCA axis 1 [plant-level live nodule biomass (+), mean ramet height (+), mean ramet diameter (+), and specific leaf mass (-)]; PCA2 = PCA axis 2 [number of live ramets per plant (+) and dead ramets per plant (+)]. Values reflect mean ± standard error.
(DOCX)

**S2 Table. Total variance explained by a principal component analysis (PCA) of alder growth traits.** The PCA was conducted with 2015 plots (n = 40) and includes: plant-level live nodule biomass (g nodule m$^{-2}$ plant$^{-1}$), mean ramet height (m), specific leaf mass (g cm$^{-2}$), mean ramet diameter (cm), and a count of live and dead ramets per plant. Eigenvalue cutoff was set at 1.
(DOCX)

**S3 Table. Descriptive statistics for alder growth traits across the region.** Statistics were calculated with the 2015 dataset (n = 40) unless stated otherwise (2014 dataset, n = 200). Significant differences between the Boundary Fire and Wickersham Dome Fire are determined at p < 0.05 level and shown in bold font. Statistics are calculated for plots in which alder was present. NODBIO = live nodule biomass (g nodule m$^{-2}$ plant$^{-1}$); SLM = specific leaf mass (mg cm$^{-2}$); Height = mean ramet height (m); MRD = mean ramet diameter (cm); LRPP = live ramets per plant; DRPP = dead ramets per plant; PCA1 = PCA axis 1 [plant-level live nodule biomass (+), mean ramet height (+), mean ramet diameter (+), and specific leaf mass (-)]; PCA2 = PCA axis 2 [number of live ramets per plant (+) and dead ramets per plant (+)]. Values reflect mean ± standard error.
(DOCX)

**S4 Table. Descriptive statistics of the alder growth traits in the Boundary Fire.** Statistics were calculated using the 2015 dataset (n = 19), unless stated otherwise (2014 dataset, n = 125). Variables significantly different (p < 0.05) across stand types are shown in bold print. Different letters among columns in the same row indicate significant differences among stand types at p < 0.05. NODBIO = live nodule biomass (g nodule m$^{-2}$ plant$^{-1}$); Height = mean ramet height (m); SLM = specific leaf mass (mg cm$^{-2}$); MRD = mean ramet diameter (cm);

LRPP = live ramets per plant; DRPP = dead ramets per plant; PCA1 = PCA axis 1 [plant-level live nodule biomass (+), mean ramet height (+), mean ramet diameter (+), and specific leaf mass (-)]; PCA2 = PCA axis 2 [number of live ramets per plant (+) and dead ramets per plant (+)]. Values reflect mean ± standard error.
(DOCX)

**S1 File.**
(ZIP)

**S2 File.**
(ZIP)

## Acknowledgments

We thank Sarah Lily and B'Elanna Rhodehamel for field assistance; Lola Oliver of the UAF Forest Soils Lab and Karl Olson of the Bonanza Creek LTER for laboratory assistance; the Institute of Arctic Biology and the Boreal Ecology Cooperative Research Unit for logistical support of fieldwork; and Gerald Frost for comments on previous drafts.

## Author Contributions

**Conceptualization:** Roger Ruess, Teresa Hollingsworth.

**Data curation:** Brian Houseman.

**Formal analysis:** Brian Houseman.

**Funding acquisition:** Roger Ruess, Teresa Hollingsworth.

**Investigation:** Brian Houseman.

**Methodology:** Brian Houseman, Roger Ruess, Teresa Hollingsworth, Dave Verbyla.

**Visualization:** Brian Houseman.

**Writing – original draft:** Brian Houseman, Roger Ruess, Teresa Hollingsworth, Dave Verbyla.

**Writing – review & editing:** Brian Houseman, Roger Ruess, Teresa Hollingsworth, Dave Verbyla.

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
