## [Decision Letter · Decision Letter 0]

8 May 2020

PONE-D-20-08364

Can Siberian alder N-fixation offset N-loss after severe fire? Quantifying post-fire Siberian alder distribution, growth, and N-fixation in boreal Alaska.

PLOS ONE

Dear Mr. Houseman,

Thank you for submitting your manuscript to PLOS ONE. After careful consideration, we feel that it has merit but does not fully meet PLOS ONE’s publication criteria as it currently stands. Therefore, we invite you to submit a revised version of the manuscript that addresses the points raised during the review process.

ACADEMIC EDITOR: Your study has good results and new findings,but not presented clearly. Please reorganize your paper according to the suggestions of the reviewer to make it more understandable for readers.

We would appreciate receiving your revised manuscript by Jun 22 2020 11:59PM. To enhance the reproducibility of your results, we recommend that if applicable you deposit your laboratory protocols in protocols.io, where a protocol can be assigned its own identifier (DOI) such that it can be cited independently in the future. For instructions see: http://journals.plos.org/plosone/s/submission-guidelines#loc-laboratory-protocols

We look forward to receiving your revised manuscript.

Kind regards,

RunGuo Zang

Academic Editor

PLOS ONE

Journal Requirements:

Please consider the concerns of the reviewer,and make revisions accordingly.

Reviewers' comments:

Reviewer's Responses to Questions

**Comments to the Author**

1. Is the manuscript technically sound, and do the data support the conclusions?

Reviewer #1: Yes

2. Has the statistical analysis been performed appropriately and rigorously? 

Reviewer #1: Yes

3. Have the authors made all data underlying the findings in their manuscript fully available?

Reviewer #1: Yes

4. Is the manuscript presented in an intelligible fashion and written in standard English?

Reviewer #1: No

5. Review Comments to the Author

Reviewer #1: Also Provided as an Attachment

Review of PONE-D-20-08364: Can Siberian alder N-fixation offset N-loss after severe fire? Quantifying post-fire Siberian alder distribution, growth, and N-fixation in boreal Alaska

Summary:

This manuscript describes a study of Siberian Alder N fixation in two burn scars near Fairbanks, AK that differ in time since last fire and encompass a range of stand types, edaphic conditions, and (in one burn scar) fire severity. The authors find that nodule-level fixation activity was substantially higher in the younger (11 yr) burn scar than the older (40 yr) burn scar, but that alder abundances were higher on average in the older burn scar leading to negligible differences in total N fixation N inputs between the two burn scars on average. One of the strongest effects on alder N fixation inputs was stand dynamics (both pre- and post-fire) within the two burn scars. The general finding is that spots with high-severity fire that converted previous spruce stands to deciduous stands (but killed off alder rhizotomaceous stems) had very low N fixation inputs, but that similar-age stands that were previously deciduous forest and received less severe burning had relatively high N fixation inputs. The authors suggest that a complex suite of soil and topography characteristics, some of which are driven by the prior tree community and by fire severity, create conditions that help dictate variation in N fixation across the study site. They then note that these differences can lead to substantial inbalances (both positive and negative) in N budgets of post-fire boreal forest stands depending on how fire and stand dynamics impact N fixation inputs.

General Comments to Authors:

In general, this paper represents a thorough study of an important issue and encompasses an impressive amount of sampling and statistical work. In general, I don’t have many technical issues with the study; however, I think the presentation of the work in the manuscript needs to be substantially improved for it to be published. As currently written, the methods and, especially, the results are sufficiently convoluted that it’s difficult to make any reasonable sense of the authors’ findings. To this end, I’ve provided both general and specific comments that I hope will help the authors improve the clarity of the manuscript.

The objectives set out at the end of the introduction don’t fit particularly well with the rest of the study. The main objective of the study focuses on the effects of fire severity, but fire severity per se is only a piece of the results that you present. You also note the specific objective of looking at alder’s influence on the N balance of your sites, but that’s not even presented in the results (only the discussion currently). There is nothing specific in these objectives about the soil properties or stand dynamics that end up being the important (and in my opinion, very interesting) drivers of various N fixation variables.

In the methods, you implement quite a wide array of different statistical tools, which I think do a good job of catering to the specific questions/data that they are used for. That said, even statistically savvy readers likely won’t be familiar with all of these tools so I would suggest a bit more (concise) explanation on what each test/model is used for and why.

The biggest issue in my mind is that the structural presentation of the results section needs substantial revision. When one looks just at the subheadings within the results, the reader is presented with, in order: N fixation, Siberian Alder growth traits, Siberian alder growth (how is this different than the previous section), Siberian Alder density, Annual stand-level N fixation input, then Modeling Siberian alder growth traits, Siberian alder density, Siberian Alder growth, and Annual plant-level N fixation input. So most sections are duplicated and different results are presented in the two subheadings with the same name – this makes it extremely hard to keep track of the information the authors are trying to convey.

From what I can tell, the first set of Results subheadings present the results of the ANOVA and multiple regression models, then the latter set focus on the results of the SEM’s. If so, this is an almost prohibitively confusing way to present the results. It’s extremely difficult for the reader to know if the ANOVA/regression results for a given variable generally agree with the SEM models, and if they don’t agree, which one is best supported, etc.?

To improve the structure of the results section, I would suggest having a single section for each general response variable you’re interested in and reporting all of the ways that you looked at that response variable together in each of those sections.

Within each results paragraphs, I appreciate the attempt to briefly summarize the results of that paragraph into digestible sentences – I would suggest making these the topic (first) sentences for each paragraph so the reader knows what the main result is immediately and is then presented with the evidence to support that.

In the discussion, the presentation of the N-budget work sort of hits the reader by surprise. This is a substantial part of your analysis, so the description in 524-536 should go in the methods section (and be fully fleshed out) and the section in 537-542 should go in the Results.

Specific Comments to Authors:

L70: You should more clearly note that the Tierney study was done is a very different ecosystem (longleaf pine savannahs of the southeast U.S.) that only has small herbaceous N fixers.

L88: “from rare (or absent)” should be changed to just “from absent” as that represents the low end of the spectrum that contrasts with “very dense.”

L89: change “unpredictable” to “variable” – predicting these distributions is largely what the ultimate goal of this work is, so hopefully it’s predictable given the right information.

L126-138: the use of the non-parametric nearest alder formula to get alder densities will probably not be intuitive to most readers (i.e., why not just count the number of alder within a given area?) so it would be good to put in a quick justification for why you chose to measure alder distributions this way.

L134: explicitly state what “nearest individual” is nearest to (I assume the plot center, but it’s currently unclear).

L143: You use metric units elsewhere in the manuscript, so it would be best to convert inches to cm here.

L162-163: It’s not clear what “systematic random order” is. How is this different from a random order?

L172: The units that you present nodule-level N fixation in are on a per-hour basis. Conventionally people only incorporate seasonal variation in N fixation rates when they scale rates up to a per-year basis. Given that this is quite different than many/most other studies, it would be good to state why you accounted for seasonal variation for per-hour rates.

L185-188: Were temperature and moisture data taken at single time points? If so, how do we know that they were done so that time of day and day-to-day variation in air temp/aridity aren’t creating superfluous variation in these data?

L195: is this a 1m buffer around the perimeter of the crown of each individual or the base of the stem? That makes a big difference in sample area.

L196: What evidence do you have that nodules tend to be uniformly distributed within these areas? To me, this actually seems pretty unlikely but if you have evidence for this it would be good to present it.

L199: indicate that this is a plot-level estimate of “per-plant” live nodule biomass (which is why you don’t need alder abundances for this metric).

L204-205: it’s best practice to list out which variables were transformed.

L227-228: it seems to me that this is where the assumptions of seasonal variation in N fixation should come into play, and how that assumption is combined with the 24-hr/day assumption is important information to include.

L280: should read “was not significantly different than” since you used a two-tailed test.

L291: This is an interesting quirk in the results – that the effect size is stronger and variation smaller for the comparison between Spruce and Deciduous stands than it is between Spruce and Mixed stands, but the Spruce-Mixed comparison is the one that is significant. It would be good to note why this is (potentially a sample size effect?).

L293: I like the summary of the results in the paragraph here but I would urge caution on statements like “as time since fire increased” given that you don’t have a whole series of forest ages.

Fig 1. This image makes it look like most plots were placed right at the edge of the burn scar or the road (at least in BF). Is this just a visual trick due to the scale or were many actually near the edge? If so, this should be noted in the methods.

Fig 2. This figure would work better as multiple figures. I would keep all of the different response variables together, but make the global comparison between BF and WDF a figure, then the stand-level comparisons within each burn scar separate figures (3 total). As is, it’s a bit of information overload which makes it difficult for the reader to decipher what the important information to glean is.

6. PLOS authors have the option to publish the peer review history of their article (what does this mean?). If published, this will include your full peer review and any attached files.

Reviewer #1: No

---

## [Author Response · Author response to Decision Letter 0]

28 Jul 2020

We have responded to each element of the decision letter by typing our response beneath the reviewer's comments (see below). A copy of our response is also attached as a Word document entitled "Response to Reviewers".

Review of PONE-D-20-08364: Can Siberian alder N-fixation offset N-loss after severe fire? Quantifying post-fire Siberian alder distribution, growth, and N-fixation in boreal Alaska

Summary: 

This manuscript describes a study of Siberian Alder N fixation in two burn scars near Fairbanks, AK that differ in time since last fire and encompass a range of stand types, edaphic conditions, and (in one burn scar) fire severity. The authors find that nodule-level fixation activity was substantially higher in the younger (11 yr) burn scar than the older (40 yr) burn scar, but that alder abundances were higher on average in the older burn scar leading to negligible differences in total N fixation N inputs between the two burn scars on average. One of the strongest effects on alder N fixation inputs was stand dynamics (both pre- and post-fire) within the two burn scars. The general finding is that spots with high-severity fire that converted previous spruce stands to deciduous stands (but killed off alder rhizotomaceous stems) had very low N fixation inputs, but that similar-age stands that were previously deciduous forest and received less severe burning had relatively high N fixation inputs. The authors suggest that a complex suite of soil and topography characteristics, some of which are driven by the prior tree community and by fire severity, create conditions that help dictate variation in N fixation across the study site. They then note that these differences can lead to substantial inbalances (both positive and negative) in N budgets of post-fire boreal forest stands depending on how fire and stand dynamics impact N fixation inputs. 

General Comments to Authors: 

In general, this paper represents a thorough study of an important issue and encompasses an impressive amount of sampling and statistical work. In general, I don’t have many technical issues with the study; however, I think the presentation of the work in the manuscript needs to be substantially improved for it to be published. As currently written, the methods and, especially, the results are sufficiently convoluted that it’s difficult to make any reasonable sense of the authors’ findings. To this end, I’ve provided both general and specific comments that I hope will help the authors improve the clarity of the manuscript. 

The objectives set out at the end of the introduction don’t fit particularly well with the rest of the study. The main objective of the study focuses on the effects of fire severity, but fire severity per se is only a piece of the results that you present. You also note the specific objective of looking at alder’s influence on the N balance of your sites, but that’s not even presented in the results (only the discussion currently). There is nothing specific in these objectives about the soil properties or stand dynamics that end up being the important (and in my opinion, very interesting) drivers of various N fixation variables.

We agree that the language used to describe our objectives was vague. We have made updates to the introduction which more accurately describe the original intent of this study. The effect of fire severity on alder density (and therefore N-input) was the main component of our hypothesis. However, we could not determine the effect of fire severity without first disentangling it from the effects of other predictor variables (soil properties, topography). We also needed to control for stand type and fire age, both of which figured prominently into our analysis. We have updated objective 1 to clarify how soil properties, topography, stand type, and fire age align with our hypothesis. Furthermore, we have changed objective 2 from an objective to a goal because doing so more clearly defines the purpose of our N-balance analysis.

We referred to “environmental characteristics” in the original text of the introduction (specifically objective 1), a term that is intended to capture topoedaphic variables (soil properties, topography). As such, our methods and results directly addressed that part of objective 1 (we measure and analyze the effect of soil properties and topographic characteristics on alder N-input). Regardless, we agree that “environmental characteristics” is not a clear term. We updated objective 1 to include our definition of environmental characteristics. 

We have moved the N-balance material out of the Discussion and into the Methods and Results sections, as suggested below. 

In the methods, you implement quite a wide array of different statistical tools, which I think do a good job of catering to the specific questions/data that they are used for. That said, even statistically savvy readers likely won’t be familiar with all of these tools so I would suggest a bit more (concise) explanation on what each test/model is used for and why. 

We have added text that describes what each statistical tool is used for and why. We have also rearranged the methods and results sections which should make the validity of each statistical tool clearer.

The biggest issue in my mind is that the structural presentation of the results section needs substantial revision. When one looks just at the subheadings within the results, the reader is presented with, in order: N fixation, Siberian Alder growth traits, Siberian alder growth (how is this different than the previous section), Siberian Alder density, Annual stand-level N fixation input, then Modeling Siberian alder growth traits, Siberian alder density, Siberian Alder growth, and Annual plant-level N fixation input. So most sections are duplicated and different results are presented in the two subheadings with the same name – this makes it extremely hard to keep track of the information the authors are trying to convey. 

We have rearranged and consolidated the results section for improved clarity.

From what I can tell, the first set of Results subheadings present the results of the ANOVA and multiple regression models, then the latter set focus on the results of the SEM’s. If so, this is an almost prohibitively confusing way to present the results. It’s extremely difficult for the reader to know if the ANOVA/regression results for a given variable generally agree with the SEM models, and if they don’t agree, which one is best supported, etc.? 

We have rearranged the presentation of the results section by response variable (as suggested below) which addresses the disparate locations of statistical results.

To improve the structure of the results section, I would suggest having a single section for each general response variable you’re interested in and reporting all of the ways that you looked at that response variable together in each of those sections. 

We have updated the results section so that each subheading is a response variable and a separate subheading was created for post-fire N-balance. The various statistical tools applied to each response variable are now summarized under a single subheading for each response variable.

Within each results paragraphs, I appreciate the attempt to briefly summarize the results of that paragraph into digestible sentences – I would suggest making these the topic (first) sentences for each paragraph so the reader knows what the main result is immediately and is then presented with the evidence to support that. 

We updated the text according to this suggestion.

In the discussion, the presentation of the N-budget work sort of hits the reader by surprise. This is a substantial part of your analysis, so the description in 524-536 should go in the methods section (and be fully fleshed out) and the section in 537-542 should go in the Results. 

Agreed. We have moved this text out of the discussion and into the methods and results sections, as specified. We added more detail, as necessary, within the methods and results descriptions. A new subheading (Post-fire N-balance) was adopted and used throughout the manuscript.

Specific Comments to Authors: 

L70: You should more clearly note that the Tierney study was done is a very different ecosystem (longleaf pine savannahs of the southeast U.S.) that only has small herbaceous N fixers. 

We have made the location of the Tierney study clearer, but we decided it was not necessary to mention the stature of herbaceous N-fixers in longleaf pine savannas. Though legumes are smaller than alder, their rates of N-fixation in this ecosystem rival that of Siberian alder in black spruce stands of our study.

L88: “from rare (or absent)” should be changed to just “from absent” as that represents the low end of the spectrum that contrasts with “very dense.”

We have changed the text accordingly.

L89: change “unpredictable” to “variable” – predicting these distributions is largely what the ultimate goal of this work is, so hopefully it’s predictable given the right information. 

We have changed the text accordingly.

L126-138: the use of the non-parametric nearest alder formula to get alder densities will probably not be intuitive to most readers (i.e., why not just count the number of alder within a given area?) so it would be good to put in a quick justification for why you chose to measure alder distributions this way.

A brief justification of a nonparametric density estimator was added to the text. On a separate note: a citation was added with this edit but tracked changes needed to be turned off in order for the citation add-in tool to update the references section of the manuscript.

L134: explicitly state what “nearest individual” is nearest to (I assume the plot center, but it’s currently unclear).

We updated the definition of “nearest individual” in this sentence.

L143: You use metric units elsewhere in the manuscript, so it would be best to convert inches to cm here. 

Agreed. We made this change in the text.

L162-163: It’s not clear what “systematic random order” is. How is this different from a random order?

We added a detailed explanation of the sampling method and removed the confusing term “systematic random order”. The intent of that term was to indicate our random selection of plots by stratum instead of by randomly selecting from among all plots regardless of stratum. Previous studies of N-fixation input by alder in Alaska were confounded by the seasonality of alder N-fixation when all plots within a stratum were sampled earlier in the summer than all plots within another stratum (Anderson and others 2004).

L172: The units that you present nodule-level N fixation in are on a per-hour basis. Conventionally people only incorporate seasonal variation in N fixation rates when they scale rates up to a per-year basis. Given that this is quite different than many/most other studies, it would be good to state why you accounted for seasonal variation for per-hour rates. 

This was a typo in the original manuscript and has been corrected to indicate that seasonal variation in N-fixation was accounted for in plant-level and stand-level N-input, but not in hourly rates (as should be the case). Additional text was added to indicate that nodule-level N-fixation rates reflect peak N-fixation (July). These new descriptions of the methods accurately reflect the methods we employed in this study.

L185-188: Were temperature and moisture data taken at single time points? If so, how do we know that they were done so that time of day and day-to-day variation in air temp/aridity aren’t creating superfluous variation in these data? 

We controlled for variation in soil moisture and temperature by time of day and day-to-day differences by sampling plots within each sampling stratum randomly throughout the 30-day sampling period, and throughout a single day. Soil moisture and temperature were not significantly correlated with day-of-year or time of day, and soil moisture and temperature did not vary among stand types within a burn scar or between burn scars. We have updated the text to indicate that soil moisture and temperature variation was controlled for based on our sampling design. 

L195: is this a 1m buffer around the perimeter of the crown of each individual or the base of the stem? That makes a big difference in sample area. 

The area of nodulation extended 1m away from the outermost ramets of an individual alder. We updated the text to make this definition clear. We incorporate the size of the area of nodulation when calculating N-fixation (from the plant to stand scale) and we discuss how variation in area of nodulation affects our results (including the effect of simply defining a limit for the purpose of sampling).

L196: What evidence do you have that nodules tend to be uniformly distributed within these areas? To me, this actually seems pretty unlikely but if you have evidence for this it would be good to present it.

We sampled hundreds of plants for N-fixation and nodule biomass and discovered that nodules were fairly uniform within 1 meter of the plant. Beyond 1 meter, nodule density and distribution became quite sparse. Because this is a personal observation, and because the term “uniformly distributed” is open to interpretation we have decided to delete this comment form the text.

L199: indicate that this is a plot-level estimate of “per-plant” live nodule biomass (which is why you don’t need alder abundances for this metric). 

We added text to indicate that plot-level estimates of live nodule biomass are on a per-plant basis.

L204-205: it’s best practice to list out which variables were transformed. 

We have added lists of transformed variables to the Statistical analysis section.

L227-228: it seems to me that this is where the assumptions of seasonal variation in N fixation should come into play, and how that assumption is combined with the 24-hr/day assumption is important information to include. 

Agreed. We have updated the text to indicate that seasonal variation assumption was accounted for in combination with the 24-hr/day assumption. The previous description of our methods was unclear on this point, and the updated description reflects the fact that our methods of calculating plant-level N-fixation input (and by extension stand-level N-fixation input) matched conventional methods for doing so.

L280: should read “was not significantly different than” since you used a two-tailed test.

We have updated the text as suggested.

L291: This is an interesting quirk in the results – that the effect size is stronger and variation smaller for the comparison between Spruce and Deciduous stands than it is between Spruce and Mixed stands, but the Spruce-Mixed comparison is the one that is significant. It would be good to note why this is (potentially a sample size effect?). 

Yes, low sample size may explain the lack of significant difference between these two stand types despite the effect size. We have added a note addressing this quirk and updated the supplemental tables to include stand type sample sizes.

L293: I like the summary of the results in the paragraph here but I would urge caution on statements like “as time since fire increased” given that you don’t have a whole series of forest ages. 

Agreed. We have updated the text accordingly.

Fig 1. This image makes it look like most plots were placed right at the edge of the burn scar or the road (at least in BF). Is this just a visual trick due to the scale or were many actually near the edge? If so, this should be noted in the methods.

Yes, many study plots were relatively close to the burn scar edge. We added a note in the Study area section describing plot locations and the lack of overlap between plots and unnatural influences.

Fig 2. This figure would work better as multiple figures. I would keep all of the different response variables together, but make the global comparison between BF and WDF a figure, then the stand-level comparisons within each burn scar separate figures (3 total). As is, it’s a bit of information overload which makes it difficult for the reader to decipher what the important information to glean is. 

We believe Fig 2 works better in tis current format because one can compare alder N input variables among stand types and across burn scars in a single figure. Therefore, we have not changed Fig 2.

---

## [Editor Report · Decision Letter 1]

7 Aug 2020

Can Siberian alder N-fixation offset N-loss after severe fire? Quantifying post-fire Siberian alder distribution, growth, and N-fixation in boreal Alaska.

PONE-D-20-08364R1

Dear Dr. Houseman,

We’re pleased to inform you that your manuscript has been judged scientifically suitable for publication and will be formally accepted for publication once it meets all outstanding technical requirements.

Kind regards,

RunGuo Zang

Academic Editor

PLOS ONE

Additional Editor Comments (optional):

accept
---

## [Editor Report · Acceptance letter]

19 Aug 2020

PONE-D-20-08364R1 

Can Siberian alder N-fixation offset N-loss after severe fire? Quantifying post-fire Siberian alder distribution, growth, and N-fixation in boreal Alaska. 

Dear Dr. Houseman:

I'm pleased to inform you that your manuscript has been deemed suitable for publication in PLOS ONE. Congratulations! Your manuscript is now with our production department. 

Kind regards, 

on behalf of

Professor RunGuo Zang 

Academic Editor

PLOS ONE